# Association between *MGMT* Enhancer Methylation and *MGMT* Promoter Methylation, MGMT Protein Expression, and Overall Survival in Glioblastoma

**DOI:** 10.3390/cells12121639

**Published:** 2023-06-15

**Authors:** Katja Zappe, Katharina Pühringer, Simon Pflug, Daniel Berger, Andreas Böhm, Sabine Spiegl-Kreinecker, Margit Cichna-Markl

**Affiliations:** 1Department of Analytical Chemistry, Faculty of Chemistry, University of Vienna, 1090 Vienna, Austria; katja.zappe@univie.ac.at (K.Z.); katharina.puehringer@univie.ac.at (K.P.); n12030136@students.meduniwien.ac.at (S.P.); office@berger-daniel.at (D.B.); andi.boehm@yahoo.de (A.B.); 2Department of Neurosurgery, Kepler University Hospital GmbH, Johannes Kepler University, 4040 Linz, Austria; sabine.spiegl-kreinecker@kepleruniklinikum.at

**Keywords:** glioblastoma, MGMT, overall survival, prognostic biomarker, enhancer, DNA methylation, enhancer methylation, biomarker, high resolution melting, pyrosequencing

## Abstract

The repair protein O6-methylguanine-DNA methyltransferase (MGMT) is regulated epigenetically, mainly by the methylation of the *MGMT* promoter. *MGMT* promoter methylation status has emerged as a prognostic and predictive biomarker for patients with newly diagnosed glioblastoma (GBM). However, a strong negative correlation between *MGMT* promoter methylation and MGMT protein expression cannot be applied as a rule for all GBM patients. In order to investigate if the DNA methylation status of *MGMT* enhancers is associated with *MGMT* promoter methylation, MGMT expression, and the overall survival (OS) of GBM patients, we established assays based on high-resolution melting analysis and pyrosequencing for one intragenic and three intergenic *MGMT* enhancers. For CpGs in an enhancer located 560 kb upstream of the *MGMT* promoter, we found a significant negative correlation between the methylation status and MGMT protein levels of GBM samples expressing MGMT. The methylation status of CpGs in the intragenic enhancer (hs696) was strongly negatively correlated with *MGMT* promoter methylation and was significantly higher in MGMT-expressing GBM samples than in MGMT-non-expressing GBM samples. Moreover, low methylation of CpGs 01–03 and CpGs 09–13 was associated with the longer OS of the GBM patients. Our findings indicate an association between *MGMT* enhancer methylation and *MGMT* promoter methylation, MGMT protein expression, and/or OS.

## 1. Introduction

Glioblastoma (GBM) accounts for approximately 45–50% of all primary malignant brain tumors in adults. GBMs have been defined as isocitrate dehydrogenase (IDH)-wild-type diffuse astrocytomas by the World Health Organization (WHO) [1]. With a median survival time of 15 months after diagnosis, prognosis of GBM is very poor. Standard therapy of newly diagnosed GBM patients consists of radiotherapy and concomitant therapy with the alkylating agent temozolomide (TMZ), followed by adjuvant treatment with TMZ [2]. TMZ causes DNA damage by the addition of methyl groups at the N7 and O6 positions of guanine and the N3 position of adenine residues [3]. However, response to TMZ critically depends on the expression of O6-methylguanine-DNA methyltransferase (MGMT), since it repairs alkylation lesions, preferentially methylation at position O6 of guanine [4].

MGMT is regulated epigenetically, mainly by the methylation status of the *MGMT* promoter [5]. DNA methylation is the most widely studied epigenetic modification. It plays a crucial role in gene regulation by affecting chromatin accessibility and thus gene transcription. Methylation of CpG dinucleotides (CpGs) in the promoter region commonly leads to transcriptional inactivation of genes [6]. The *MGMT* promoter contains a CpG island which consists of 98 CpGs and extends into the non-coding exon 1 (CpGs 74–83; NCBI Reference Sequence: NG_052673.1) [7]. Within the CpG island, a minimal promoter (88 bp, CpGs 50–62; −173 to −86 according to NCBI exon 1 coordinates) and an enhancer region (59 bp, CpGs 82–87; +40 to +98) have been identified [8]. Hypermethylation of the *MGMT* promoter, resulting in transcriptional silencing, has been associated with survival benefits from TMZ therapy [9]. Since this finding was confirmed in clinical trials [10,11,12], *MGMT* promoter methylation status has emerged as a predictive biomarker for the response to TMZ, in particular in the population of elderly patients with newly diagnosed GBM [13]. In addition, several studies have demonstrated the suitability of *MGMT* promoter methylation as a prognostic biomarker for GBM [14,15].

However, growing evidence suggests that a strong negative correlation between *MGMT* promoter methylation and MGMT protein expression does not apply to all GBM patients. There are tumors that express MGMT in spite of *MGMT* promoter methylation, and others that do not show MGMT expression although the promoter is unmethylated [16]. Promoter methylation is still assumed to play a major role in *MGMT* silencing. However, other mechanisms seem to affect the correlation between *MGMT* promoter methylation and MGMT protein expression as well [17], and under certain circumstances, even the overrule influence of *MGMT* promoter methylation.

We hypothesize that DNA methylation of *MGMT* enhancers is one of the factors playing a role in regulating MGMT expression in addition to *MGMT* promoter methylation. Enhancers are DNA regulatory elements that precisely control spatiotemporal patterns of gene expression [18]. In contrast to promoters, distal enhancers may be located upstream or downstream and even at a far distance from the transcription start site (TSS) of their target gene(s). Increasing evidence suggests that distal enhancers interact with promoters through long-range interactions via chromatin looping [19]. In general, enhancer activity may be affected by alterations in the enhancer sequence, including mutations and single nucleotide polymorphisms (SNPs), that alter chromatin accessibility and/or transcription factor binding [20]. In addition, the DNA methylation status of enhancers may have an impact on their activity. Enhancer regions are frequently depleted of CpGs and characterized by low DNA methylation levels [21]. However, by analyzing datasets for 58 cell types, Aran et al. found out that enhancers were gradually methylated and that enhancer methylation correlated stronger with gene expression than promoter methylation [22]. Very recently, Kreibich et al. proposed that DNA methylation of CpGs in enhancers plays a role in controlling the binding of transcription factors [23]. Studies indicate that aberrant enhancer methylation is a frequent event in various cancer types [24,25,26,27], including glioblastoma [28].

In the present study, we determined the methylation status of CpGs in distal *MGMT* enhancers in samples from 38 IDH-wildtype GBM, one IDH mutated GBM, one gliosarcoma, and for the commercial cell line T98G. We investigated if enhancer methylation is associated with *MGMT* promoter methylation, MGMT protein expression, and/or OS. We selected one intragenic (hs696) and two intergenic (hs737, hs699) enhancers from the VISTA Enhancer Browser [29] and an intergenic enhancer that was recently identified by Chen et al. [30]. For DNA methylation analysis of the *MGMT* promoter, we used a primer set from the literature [31], targeting CpGs 72–83 of the promoter. The DNA methylation status of the enhancers was determined by using methods developed in-house. We amplified the target regions by the polymerase chain reaction (PCR) and subjected PCR products to high-resolution melting (HRM) analysis and pyrosequencing (PSQ). Since HRM analysis provides information on the average methylation status across all CpGs in the PCR product, it was used for screening. In addition, HRM analysis was applied to obtain information on the occurrence of specific methylation patterns such as monoallelic methylation. The methylation status of individual CpGs was obtained by PSQ. 

## 2. Materials and Methods

### 2.1. Samples and Cell Culturing

The sample set consisted of primary human tumor cell lines established from 40 glioma patients who underwent surgery between 2001 and 2020 at the Department of Neurosurgery, Kepler University Hospital, Neuromed Campus, as described previously [32], and the commercial GBM cell line T98G (ATCC, Manassas, VA, USA). The study was approved by the local Ethics Commission of the Faculty of Medicine at the Johannes Kepler University Linz (application number E-39-15). All patients signed a written informed consent form. Thirty-eight patients suffered from IDH-wild-type GBM (mean age at surgery 62.58 years; median 63.50 years), one patient (32 years) was diagnosed with IDH1-mutated (R132H) GBM, and one patient (43 years) was diagnosed with primary gliosarcoma (GS). Tumor-derived cell cultures were diagnosed according to the current version of the WHO classification of CNS tumors before 2022, therefore using the terms GBM (IDH wt) and GBM IDH mut. Overall survival was defined as the period between the date of surgery and death.

The cell lines were cultured in RPMI-1640, 7% fetal calf serum (FCS), and 1% glutamine without antibiotics (all Sigma-Aldrich, Schnelldorf, Germany) at 37 °C in a humidified 5% CO_2_ incubator and harvested before reaching confluence between passages 2 and 6. The cell pellets were stored at −80 °C until DNA extraction. Relative MGMT protein expression, given as a ratio to MGMT overexpressing glioblastoma cell line GL80, was determined previously by Western blot analysis [33]. 

### 2.2. DNA Extraction and Bisulfite Conversion

Genomic DNA was isolated using the QIAamp DNA Blood Mini Kit (Qiagen, Hilden, Germany) following the manufacturer’s protocol for cultured cells and quantified with a Qubit dsDNA BR Assay Kit using the Qubit 4 instrument (Thermo Fisher Scientific, Vienna, Austria). The DNA extracts were stored at −20 °C until PCR or bisulfite conversion. For bisulfite conversion of unmethylated cytosines, a EpiTect Fast Bisulfite Conversion Kit (Qiagen, Germany) was used according to the manufacturer’s protocol. The converted DNA was quantified with a Qubit ssDNA Assay Kit (Thermo Fisher Scientific) and stored at −20 °C until PCR.

### 2.3. Primer Design and PCR Conditions for DNA Methylation Analysis

Nucleotide sequences were retrieved from the National Center for Biotechnology Information (NCBI) database [34]. For the *MGMT* promoter, coordinate definition and CpG numbering from the University of California at Santa Cruz (UCSC) Genome Browser [35] was used as suggested by Wick et al. [7] (Genbank GRCh38.p13 chr10 NC_000010.11:129466685–129467446, 98 CpGs). Sequences of two *MGMT* enhancers, enhancer 1 (hs737, NC_000010.11:128568604–128569741, 27 CpGs) and enhancer 3 (hs699, NC_000010.11:129033193–129034911, 33 CpGs), and one intragenic *MGMT* enhancer, enhancer 4 (hs696, NC_000010.11:129605804–129607046, 26 CpGs), were taken following the coordinate definition given in the VISTA Enhancer Browser [36]. The coordinates for the sequence of the *MGMT* enhancer, enhancer 2, (NC_000010.11:128906630–128909942, 46 CpGs), were taken from Chen et al. [30], including deletion regions “Del 1” (NC_000010.11:128906630–128908285, CpGs 1–25) and “Del 2” (NC_000010.11: 128908286–128909942, CpGs 26–46). The enhancers are numbered according to their position on chromosome 10; the CpGs are numbered according to their position in the respective enhancer (Figure 1). 

The DNA methylation status of CpGs 72–83 of the *MGMT* promoter was determined by using a primer set from the literature [31]. In total, ten assays targeting enhancer regions were developed in-house using PyroMark Assay Design Software 2.0.1.15 (Qiagen), with the primer sequences given in Table 1. 

For each primer set, the primer concentration, PCR mix as well as annealing and elongation temperature and time were optimized by using bisulfite-converted human methylated and non-methylated DNA (Zymo Research, Irvine, CA, USA) and a 50% mixture of both DNA standards. Each reaction was performed in a total volume of 20 μL, consisting of 1× PCR mix including EvaGreen HRM dye, forward and reverse primer, and 5 ng of bisulfite converted DNA. PCR was carried out using either a QuantStudio 5 instrument (Thermo Fisher Scientific) or a Rotor-Gene Q instrument with a 72-well rotor (Qiagen) for assay optimization and the Rotor-Gene Q instrument for sample analysis. The optimized PCR conditions are listed in Appendix A. For all assays, the following HRM program was applied directly after final elongation: strand separation for 1 min at 95 °C, strand hybridization for 1 min at 40 °C and HRM with a ramp from 65 °C to 95 °C with 0.1 °C/hold (2 s) and gain optimization (70% before melt). 

Each PCR run included bisulfite-converted human non-methylated and methylated DNA, 25%, 50%, and 75% mixtures thereof, and a no template control (2 μL nuclease-free H_2_O). All samples were analyzed in two independent PCR runs, in two replicates per PCR.

The identity, quality, and yield of PCR products obtained for standards were assessed by gel electrophoresis (3% agarose gel in 1× TBE (Tris-borate-EDTA) buffer. The gel was post-stained with 3× GelRed (Biotium, Fremont, CA, USA), and bands were visualized with a UVT-20 M transilluminator (Herolab, Wiesloch, Germany).

### 2.4. PSQ of PCR Products 

PSQ was performed using the PyroMark Q24 Vacuum Workstation and PyroMark Q24 Advanced instrument with PyroMark Q24 Advanced Accessories, PyroMark Q24 Advanced CpG Reagents (all Qiagen), and Sepharose High-Performance beads (GE Healthcare; Thermo Fisher Scientific) according to the manufacturer´s instructions. 

If necessary, dispensation orders were adapted, e.g., to overcome sequencing frameshifts (Appendix A). The DNA immobilization reaction was optimized in a range from 48–160 μL. The 120 μL immobilization reaction set-up contained 22.5 μL biotinylated PCR product (pool of both wells from one PCR), 1.5 μL Streptavidin Sepharose High-Performance beads (GE Healthcare,), 60 μL PyroMark Binding Buffer, and 36 μL of high-purity water (18.2 MΩ cm, ELGA PURELAB Ultra MK 2, Veolia, Celle, Germany). For the other immobilization reaction volumes, all of the components were up- or downscaled accordingly. The final immobilization reaction volume was 120 μL for assays targeting enhancers 1, 3, and 4. For assays A, B, and D of enhancer 2, the immobilization reaction volume was 160 μL, and for assay C, the immobilization reaction volume was 48 μL. DNA immobilization was performed under agitation for 10 min at 1400 rpm. The captured PCR product was denatured, washed, and the biotinylated strand was transferred into a PyroMark Q24 Plate containing 20 μL of 0.375 μM sequencing primer in PyroMark Annealing Buffer. The plate was heated at 80 °C for 5 min and then transferred into the instrument, holding the PyroMark Q24 Cartridge loaded according to pre-run information provided by PyroMark Q24 Advanced software 3.0.0 (Qiagen). A representative pyrogram for each primer set is shown in Appendix A.

### 2.5. Data Analysis and Statistics

The amplification and melting curves obtained by PCR-HRM were assessed and exported using Rotor-Gene Q Series Software 2.3.1 (Qiagen). The PSQ data were evaluated and exported with PyroMark Q24 Advanced software 3.0.0 (Qiagen). The exported data were analyzed and are presented graphically using R version 3.6.2 [37]. The R-packages used, including corrplot, ggplot2, polynom, rstatix, survival, and survminer, are listed in the Appendix A R-packages.

For HRM analysis, derivative melting curves were calculated from normalized melting curves by applying Savitzky–Golay filtering for third-degree polynomials. DNA methylation levels obtained by PSQ ≤ 5.00% (lower limit of quantification, LLOQ) and ≥95.00% (upper limit of quantification, ULOQ) were substituted with default values, namely 2.50% and 97.50%, respectively [38].

One-way ANOVA (analysis of variance) followed by a post hoc *t*-test corrected for multiple testing by Holm’s *p*-value adjustment was applied to test for significant differences between the groups. Groups consisting of only one member were excluded from testing. A scatterplot and Pearson´s correlation coefficient were used to assess the relationship between two quantitative variables. A Kaplan–Meier estimator with a log-rank test was used to analyze the survival data. *p*-values ≤ 0.05 were considered statistically significant.

## 3. Results

### 3.1. MGMT Promoter and Enhancer Methylation

Samples from 38 IDH-wild-type GBM (GBM01–38), 1 IDH-mutated GBM (GBMm01), and 1 gliosarcoma (GS01) patient and the commercial GBM cell line T98G were subjected to DNA methylation analysis of the *MGMT* promoter and 4 enhancers by HRM analysis and PSQ. Enhancers 1–3 are located upstream of the *MGMT* gene (Figure 1), and enhancer 4 is part of *MGMT* intron 2. CpGs 82–83 of the *MGMT* promoter are located in a 59 bp intrapromoter enhancer.

#### 3.1.1. Promoter Methylation

HRM curves for 19 GBM samples (GBM01, GBM10–22, GBM24, GBM33, and GBM36–38) and GBMm01 overlapped with the HRM curve for the non-methylated DNA standard (Figure 2a). In addition, the HRM results indicated that in none of the samples were CpGs 72–83 completely methylated. Specific melting curves with multiple distinct melting transitions were obtained for GBM06, GBM09, GBM23, GBM29, and T98G (Figure 2b). The PCR product of T98G had two sharp melting transitions like the PCR product obtained for the 50% standard, prepared by mixing non-methylated and completely methylated DNA strands in a ratio of 50:50 (m/m). For GBM09 and GBM29, a small shoulder in the negative derivative of the normalized melting curves suggested the co-occurrence of alleles showing low heterogenous methylation and alleles with higher mosaic methylation. For GBM06 and GBM23, the melting transition at about 75 °C was, however, not intense enough to assess whether these alleles were non-methylated or lowly methylated.

The methylation levels for the individual CpGs obtained by PSQ are shown in Figure 3. The mean methylation status of CpGs 72–83 of all glioma samples with the methylated *MGMT* promoter was 64.2%. Three samples (GBM05, GBM29, and GBM31) showed low methylation (mean < 25.0%), seven samples (GBM03, GBM08–09, GBM23, GBM30, GBM32, and T98G) showed intermediate (mean between 25.0–75.0%), and eleven samples (GBM02, GBM04, GBM06–07, GBM25–28, GBM34–35, and GS01) showed high (mean > 75.0%) methylation. At least one CpG with a methylation status > 25.0% was identified in each of the three samples with low promoter methylation status (GBM05: CpG 80; GBM29: CpGs 79–83; and GBM31: CpGs 77, 79). In most samples with the methylated *MGMT* promoter, the twelve CpGs were instead heterogeneously methylated (SD 6.6–35.6%), with the exception of GBM05, GBM26, and T98G (SD ≤ 6.5%). For most samples, no significant difference was found between the methylation status of CpGs 82–83, located in the intrapromoter enhancer, and that of CpGs 72–81 ((*p* ≥ 0.056). Only in four samples (GBM23, GBM29–30, and GBM32) did CpGs 82–83 show a higher methylation status than CpGs 72–81 (*p* ≤ 0.016).

#### 3.1.2. Enhancer 1 Methylation

Screening the methylation status of CpGs 12–19 in enhancer 1 by HRM analysis suggested that the target region was methylated in all samples except GBM13 (Figure 4a). In addition, HRM analysis indicated that in none of the samples was the target region completely methylated. Multiple distinct melting transitions were obtained for three samples, including two GBM samples with a non-methylated *MGMT* promoter (GBM12, GBM24) and the commercial cell line T98G. HRM results hint at the presence of alleles with low heterogeneous methylation in addition to alleles showing mosaic methylation. The negative derivative of normalized HRM curves for GBM12 was similar to that obtained for the 25% standard, hinting at the transition of completely methylated alleles (Figure 4a). 

The PSQ data revealed that even in GBM13, one individual CpG (CpG 18) was slightly (3.8%) methylated (Figure 3). Low methylation (mean < 25.0%) was observed for eight samples (GBM02, GBM04, GBM25, GBM28, GBM30, GBM32, GBM34, and GS01) with a methylated *MGMT* promoter and ten samples (GBM01, GBM10–11, GBM13–14, GBM16–18, GBM21, and GBM33) with a non-methylated *MGMT* promoter. High methylation of the target region was found for three samples (GBM03, GBM29, and GBM31) with a methylated *MGMT* promoter and five samples (GBM19, GBM36–38, and GBMm01) with a non-methylated *MGMT* promoter.

The mean methylation status of CpGs 12–19 of all glioma samples was 40.0%. In 35 samples, the target region was methylated rather heterogeneously (SD 5.6–28.9%); in 6 samples, it was methylated homogenously (SD ≤ 4.9%).

#### 3.1.3. Enhancer 2 Methylation

In total, 19 CpGs of enhancer 2 were targeted. However, four assays (A–D) had to be applied for determining the methylation levels of CpGs 05–08, CpGs 11–18, CpGs 24–27, and CpGs 37–39, respectively. Since amplification of the target regions turned out to be challenging, a special PCR mix that is not optimized for HRM applications had to be used. Due to the limited sample amount, samples GBM24, GBM36, and GBM37 were omitted from the analysis of CpGs 24–27 and CpGs 37–39.

GBM12 was the only sample resulting in two melting transitions, originating from non-methylated and completely methylated alleles, in all four target regions of enhancer 2 (Figure 4b shown for assay B). The HRM curves obtained for CpGs 05–08, CpGs 11–18, and CpGs 37–39 overlapped with those for the 50% standard, and the HRM curves for CpGs 24–27 overlapped with that of the 25% standard. In addition, the HRM data suggested CpGs 24–27 to be unmethylated in 19 GBM samples and T98G; CpGs 37–39 to be unmethylated in GBM02 and GBM09; CpGs 11–18 to be completely methylated in GBM03, GBM25, and GBM38; and CpGs 37–39 to be completely methylated in GBM03, GBM26, and GBM38.

PSQ revealed that CpGs 24–27 were unmethylated in nine samples (GBM02, GBM04, GBM06, GBM10, GBM13, GBM20–21, GBM30, and GBM32) and CpGs 37–39 in GBM02. The mean methylation of CpGs 05–08, CpGs 11–14, and CpGs 37–39 of all glioma samples was 54.1%, 49.0%, and 50.9%, respectively, indicating that the three regions showed similar methylation levels (Figure 3). In all three regions, most samples with (CpGs 05–08: GBM04–06, GBM08–09, GBM28–29, GBM31, GBM34–35, and T98G; CpGs 11–14: GBM05–09, GBM26, GBM28, GBM30–31, GBM34–35, GS01, and T98G; and CpGs 37–39: GBM04–06, GBM08, GBM25 GBM28–31, GBM34–35, GS01, and T98G) and without (CpGs 05–08: GBM01, GBM10–12, GBM14–22, GBM24, GBM33 GBM36–37, and GBMm01; CpGs 11–14: GBM10–11, GBM14–22, GBM24, GBM33, GBM36–37, and GBMm01; CpGs 37–39: GBM01, GBM10–12, GBM14–15, GBM17–18, GBM20–22, and GBMm01) promoter methylation showed an intermediate mean methylation level. CpGs 15–18 (mean methylation 73.4%) were significantly higher and CpGs 24–27 (mean methylation 12.5%) were significantly lower methylated than CpGs 05–08, CpGs 11–14, and CpGs 37–39 (*p* < 0.001). The mean methylation of CpGs 15–18 ranged from 58.3–87.0%, with the exception of GBM12 (21.9%) and GBM29 (11.6%). In none of the 18 samples showing rather heterogeneous methylation of CpGs 24–27 was a distinct CpG found to be unmethylated. 

#### 3.1.4. Enhancer 3 Methylation

The assay allowed for DNA methylation analysis of eight CpGs (CpGs 15–22) of enhancer 3. According to the HRM analysis, GBM28 was the only sample in which the target region was unmethylated (Figure 4c). For the PCR products of five GBM samples, multiple distinct melting transitions were obtained. GBM05–06, GBM11, and GBM34 seemed to contain alleles showing low heterogeneous methylation in addition to alleles with higher mosaic methylation (Figure 4c). The melting curve obtained for GBM12 was identical to that of the 50% standard, indicating the presence of non-methylated and completely methylated alleles in a ratio of 50:50 (m/m). 

In 13 samples (GBM02, GBM04–06, GBM08, GBM25–26, GBM28, GBM30–32, GBM35, and GS01) with a methylated *MGMT* promoter and nine samples (GBM01, GBM13–14, GBM16–18, GBM21–22, and GBM33) with a non-methylated *MGMT* promoter, the targeted CpGs showed low methylation. Intermediate methylation was obtained for seven samples (GBM03, GBM07, GBM09, GBM23, GBM27, GBM29, and GBM34) with a methylated promoter and five samples (GBM10–12 and GBM19–20) with a non-methylated promoter. High methylation was only found for six samples (GBM15, GBM24, GBM36–38, and GBMm01) in which the *MGMT* promoter was non-methylated, as well as for T98G, showing monoallelic promoter methylation. In most samples, the target region was methylated heterogeneously (SD from 3.6% to 29.6%), with the exception of GBM05, GBM06, and GBM12 (SD 1.8–3.6%).

#### 3.1.5. Enhancer 4 Methylation

Four assays (A–D) had to be applied for DNA methylation analysis of CpGs 01–03, CpGs 07–08, CpGs 09–13, and CpGs 19–22 in enhancer 4, respectively. Due to the limited sample amount, GBM24 and GBM37 were only subjected to analysis of CpGs 09–13 and CpGs 19–22. GBM36 was only analyzed for the DNA methylation status of CpGs 09–13. 

From the shape of HRM curves for the commercial non-methylated standard, we concluded that the CpGs targeted were not unmethylated but showed partial methylation (Figure 4d), which was confirmed by the PSQ data. A single sharp melting transition at the lowest Tm indicated that CpGs 01–03 were not methylated in samples GBM02 and GBM32. Complete methylation was exclusively found in promoter unmethylated samples for CpGs 01–03 (GBM11, GBM13, GBM17–18, and GBM22), for CpGs 07–13 (GBM10–11, GBM17, GBM36–38, and GBMm01; Figure 4d), and for CpGs 19–22 (GBM11, GBM13, GBM17–18, GBM22, GBM37–38, and GBMm01). Distinct melting transitions obtained for GBM12 and T98G hint at monoallelic methylation in the four target regions of enhancer 4.

PSQ analysis revealed that CpGs 19–22 were significantly more highly methylated (mean methylation of all samples 70.2%) than CpGs 01–03 (49.4%, *p* = 0.003), CpGs 07–08 (54.6%, *p* = 0.036), and CpGs 09–13 (53.3%, *p* = 0.006) (Figure 3). The mean methylation of CpGs 01–03, CpGs 07–13, and CpGs 19–22 ranged from 3.0–93.2%, 7.6–87.6%, and 11.2–95.8%, respectively. CpGs 01–03 (SD 2.1–40.3%), CpGs 07–13 (SD 5.3–32.0%), and CpGs 19–22 (SD 4.3–29.6%) were methylated rather heterogeneously. 

In seven samples (GBM02, GBM07, GBM27, and GBM29–32) with a methylated *MGMT* promoter but in none of the samples with a non-methylated promoter, CpGs 01–03 showed low methylation. On the other hand, seven samples (GBM11, GBM13–14, GBM17–18, GBM22, and GBM38) with a non-methylated *MGMT* promoter but none of the samples with a methylated promoter exhibited high methylation of CpGs 01–03. Low methylation of CpGs 07–08 and CpGs 09–13 was also exclusively found for samples with a methylated *MGMT* promoter (CpGs 07–08: GBM02, GBM04, GBM07, GBM23, GBM27, GBM29–30, and GBM32; CpGs 09–13: GBM02, GBM04, GBM08, GBM29, and GBM32), and there was high methylation exclusively for samples with a non-methylated *MGMT* promoter (CpGs 07–08: GBM10–11, GBM13–14, GBM17–18, GBM21–22, GBM33, GBM38, and GBMm01; CpGs 09–13: GBM10–11, GBM13, GBM17–18, GBM21–22, GBM24, GBM36–38, and GBMm01). For CpGs 19–22, 14 samples without (GBM10–11, GBM13–14, GBM16–18, GBM21–22, GBM24, GBM33, GBM37–38, and GBMm01), but also five samples with (GBM08–09, GBM23, GBM31, and T98G) were highly methylated.

### 3.2. Association between the MGMT Promoter and/or Enhancer Methylation

Next, we searched for correlations between the methylation status of CpGs located in the same but also in different regulatory elements (promoter/enhancer, two different enhancers). Correlation analysis was performed by including all IDH-wildtype GBM samples (GBM01–38) (Figure 5a), but we also stratified the samples by their *MGMT* promoter methylation status (Figure 5b,c).

The strongest positive correlations were found between the methylation status of CpGs in the *MGMT* promoter when all IDH-wild-type GBM samples were included (Figure 5a). Positive correlations, albeit generally weaker, were also obtained for samples with methylated *MGMT* promoters (Figure 5c). 

Strong positive correlations were also found between the methylation status of CpGs located in enhancer 1 (Figure 5). For samples with an unmethylated *MGMT* promoter, strong correlations were found between all individual CpGs (Figure 5b). In the case of promoter-methylated samples, the correlations between CpGs 12–16 were stronger than the correlations between the other CpGs targeted in enhancer 1 (Figure 5c). For enhancer 2, strong positive correlations were predominantly found between the methylation levels of CpGs 15–18, as well as between those of CpGs 37–39, independent of whether the samples stratified by their *MGMT* promoter methylation status (Figure 5b,c) or not (Figure 5a). In general, the methylation levels of CpGs targeted in enhancer 3 were also positively correlated with each other. The correlations were slightly stronger in samples with unmethylated *MGMT* promoters (Figure 5b) compared to those with methylated *MGMT* promoters (Figure 5c). The methylation levels of the CpGs located in enhancer 4 also showed positive correlations with each other. The highest number of correlations was found when all IDH-wild-type GBM samples were included in the correlation analysis (Figure 5a); the lowest number of correlations was found when the correlation analysis was restricted to samples with a methylated *MGMT* promoter (Figure 5c). 

By searching for correlations between the methylation levels of CpGs located in different regulatory elements, we found negative correlations between almost all CpGs in the *MGMT* promoter and almost all CpGs in enhancer 4, but only when the samples were not stratified by their *MGMT* promoter methylation status (Figure 5a). Among all of the CpGs targeted in enhancer 4, CpG 03 and CpG 09 showed the strongest negative correlations with the CpGs in the *MGMT* promoter. In general, the CpGs targeted in the other enhancers did not correlate with the CpGs in the promoter.

We also found positive correlations between the methylation levels of the CpGs located in different enhancers. The strongest correlations were obtained between the methylation levels of the CpGs located in enhancer 1 and enhancer 3 for samples with an unmethylated *MGMT* promoter (Figure 5b).

### 3.3. Association of the MGMT Promoter and Enhancer Methylation with MGMT Protein Expression

MGMT protein expression was only detected for samples with an unmethylated *MGMT* promoter, as well as for the commercial cell line T98G (Figure 3). GBM01 was the only sample that did not express MGMT although the *MGMT* promoter was unmethylated (Figure 3). The enhancer methylation levels for GBM01 were similar to those determined for samples with an unmethylated *MGMT* promoter that did express the MGMT protein. We therefore assume, that MGMT expression in GBM01 was not impaired by DNA methylation and excluded GBM01 from further statistical analyses. In general, statistical analyses were restricted to IDH-wild-type GBM patients (GBM02-38), because for IDH-mutated GBM patients, gliosarcoma patients, and commercial GBM cell lines, only one representative of each was available.

In the MGMT-non-expressing GBM samples, *MGMT* promoter methylation was significantly higher than in MGMT-expressing GBM samples (Figure 6a). Significant differences were found for all individual CpGs as well as for the mean methylation of CpGs 72–83 (*p* < 0.001). In contrast, the methylation levels of the CpGs targeted in enhancer 3 (Figure 6b) and enhancer 4 (Figure 6c) were higher in the MGMT-expressing GBM samples than in the non-expressing GBM samples. For enhancer 3, we found significant differences for four individual CpGs (CpG 17, 18, 21, and 22) and for the mean methylation of all CpGs targeted (*p* ≤ 0.042). In the case of enhancer 4, significant differences (*p* < 0.001) were obtained for the mean methylation of the CpGs targeted by one and the same assay (assay A–D, respectively). In addition, each individual CpG of enhancer 4 was significantly more highly methylated in the MGMT-expressing samples compared to the non-expressing samples (A–C: *p* ≤ 0.002 and D: *p* ≤ 0.013). 

The MGMT-expressing and -non-expressing samples did not differ significantly in their enhancer 1 and enhancer 2 methylation levels. Significant differences were neither found for the mean methylation of the regions targeted by one and the same assay nor for individual CpGs (Figure 6d,e). However, for the GBM samples expressing MGMT, we found a significant negative correlation between the mean methylation of all CpGs targeted in enhancer 2 and MGMT protein expression (*r* = −0.59, *p* = 0.021). The mean methylation of CpGs 05–08 (*r* = −0.64, *p* = 0.004), CpGs 11–14 (*r* = −0.72, *p* = 0.001), and CpGs 37–39 (*r* =−0.58, *p* = 0.025) were also significantly negatively correlated with MGMT expression. In addition, a significant negative correlation was found for several individual CpGs (CpG 05, 06, 07, 08 (*r* = −0.50–−0.66, *p* ≤ 0.034), CpG 11, 12, 13, 14 (*r* = −0.55–−0.77, *p* ≤ 0.019), and CpG 37 and 39 (*r* = −0.54–−0.63, *p* ≤ 0.039)) (Figure 6f).

### 3.4. Association between the MGMT Promoter and Enhancer Methylation and Overall Survival

The *MGMT* promoter was found to be either unmethylated (methylation status of all CpGs ≤ LLOQ) or methylated (mean methylation ≥ 9.6%). Among the 37 IDH-wildtype GBM patients, OS was significantly higher for patients with a methylated *MGMT* promoter than for those with an unmethylated *MGMT* promoter (Figure 7a). For patients with a methylated *MGMT* promoter, the methylation levels of CpGs 75 (*r* = 0.49, *p* = 0.039), 78 (*r* = 0.56, *p* = 0.016), and 80 (*r* = 0.57, *p* = 0.014) were positively correlated with OS. 

For the mean methylation of CpGs 01–03, CpGs 07–08, and CpGs 09–13 in enhancer 4, a cut-off of 55% turned out to be suitable for enhancer methylation, as shown for assay C in Figure 7b. Patients in which CpGs 01–03 were methylated to a lower degree had a longer OS than patients with a higher methylation status (median for <55% = 17.34 (n = 18), ≥55% = 9.88 (n = 16), *p* = 0.005). This also holds true for CpGs 9–13 (median for <55% = 14.50 (n = 22), ≥55% = 9.27 (n = 15), *p* = 0.019). Patients with low mean methylation of CpGs 01–03, CpGs 07–08, and CpGs 09–13 (assays A, B, and C) had longer OS (median for <55% = 14.65 (n = 20), ≥55% = 9.88 (n = 14), *p* = 0.011) than patients with high mean methylation. 

The methylation levels of the CpGs in enhancer 4 did not significantly correlate with OS, neither for the samples with a methylation status < 55% or a methylation status ≥ 55% (*p* ≥ 0.068). However, OS prediction based on CpGs 01–03 (assay A), CpGs 09–13 (assay C), or CpGs 01–03, CpGs 07–08 and CpGs 09–13 (assays A, B, and C) was more precise for patients with lower OS than prediction based on the promoter region (CpGs 72–83) (Figure 7a,b). This also holds true when higher cut-offs (10–25%) were used for promoter methylation. The methylation status of CpGs 19–22 (assay D) did not allow for distinguishing between patients with shorter and longer OS (*p* ≥ 0.196). 

The methylation status of the CpGs targeted in enhancers 1–3 did not allow for OS prediction using cut-offs ranging from 5–90% for enhancer methylation (Figure 7c–e, *p* ≥ 0.289). The methylation status did not correlate with OS nor when the GBM patients were stratified by their *MGMT* promoter methylation status or not (*p* ≥ 0.053).

## 4. Discussion

We determined the methylation status of 61 CpGs, with 12 CpGs being located in the *MGMT* promoter, and 49 CpGs being located in enhancers that had already been associated with the *MGMT* gene. Primer sequences for targeting 12 (CpGs 72–83) out of 98 CpGs in the *MGMT* promoter were taken from the literature [31]. CpGs 82–83 belong to a 59 bp intrapromoter enhancer. The methods for DNA methylation analysis of the enhancers were developed in-house. Enhancers 1–3 are intergenic enhancers, located upstream of the *MGMT* gene. Enhancer 4 is an intragenic enhancer, located in the *MGMT* intron 2. The assay for enhancer 1 allowed for determination of the DNA methylation status of 8 (CpGs 12–19) out of the 27 CpGs. The development of four assays (A–D) made it possible to analyze 19 (CpGs 05–08, CpGs 11–18, CpGs 24–27, and CpGs 37–39) out of the 46 CpGs of enhancer 2, respectively. The assay developed for enhancer 3 targeted 8 (CpGs 15–22) out of 33 CpGs. A total of 14 (CpGs 01–03, CpGs 07–08, CpGs 09–13, and CpGs 19–22) out of the 26 CpGs of enhancer 4 could be analyzed by the applying four assays (A–D), respectively. 

Our strategy was to amplify the target regions by PCR and to subject the PCR products to HRM analysis and subsequently to PSQ. The potential of analyzing the PCR products after HRM directly by PSQ has already been demonstrated not only for DNA methylation analysis [40], but also for SNP genotyping [41]. By providing information on the average methylation status across all CpGs in the PCR product, HRM was primarily applied for screening for non-methylated and completely methylated samples. In addition, the negative derivative of the normalized HRM curves yielded information on the occurrence of specific methylation patterns such as monoallelic methylation. In the case of monoallelic methylation, two sharp melting transitions are obtained, as for standard mixtures consisting of non-methylated and completely methylated DNA strands. Information on monoallelic methylation is of interest because it is a key mechanism of monoallelic expression, with the unmethylated allele commonly being expressed and the methylated one being silenced [42]. In contrast to HRM analysis, PSQ does not provide information on monoallelic methylation, with the methylation status for individual CpGs being the average methylation across all alleles in the sample. 

We found monoallelic methylation of the *MGMT* promoter for the commercial GBM cell line T98G. The negative derivative of normalized HRM curves overlapped with that for the 50% standard, consisting of non-methylated and completely methylated DNA strands in a ratio of 50:50 (m/m). Our finding that the *MGMT* promoter shows monoallelic methylation explains why MGMT is expressed in T98G although the promoter is highly methylated. However, under certain circumstances, protein expression may be impaired in spite of the presence of an unmethylated allele. By investigating the methylation patterns of the *MGMT* promoter in GBM samples, Kristensen et al. observed monoallelic methylation in a number of samples [43]. Among them, several did express MGMT, as expected. However, some of these samples did not express MGMT, most probably due to hemizygous deletion of the *MGMT* locus, a frequent event in GBM [43]. 

For T98G, we found monoallelic methylation not only for the *MGMT* promoter but also for the enhancer 1 and enhancer 4 regions. In the case of sample GBM12, we detected monoallelic methylation for CpGs 05–08, CpGs 11–18, CpGs 24–27, and CpGs 37–39 of enhancer 2 and CpGs 01–03, CpGs 07–13, and CpGs 19–22 of enhancer 4. In contrast with T98G, the *MGMT* promoter was unmethylated in GBM12. GBM12 was the only sample for which multiple sharp melting transitions were obtained for each of the four enhancers. GBM12 showed the third highest MGMT expression of all samples analyzed. 

Several studies have already been performed aimed at identifying individual CpGs in the *MGMT* promoter that are most applicable as potential biomarkers for the overall survival of GBM patients [7]. By determining the methylation status of 60 CpGs in 54 GBM samples, Everhard et al. found hypermethylation of six single CpGs (CpGs 27, 32, 73, 75, 79, and 80) and two CpG regions (CpGs 32–33 and CpGs 72–83) to be associated with low MGMT gene expression [44]. By using the HM-450K BeadChip, Bady et al. analyzed CpGs 17–20, 22, 25, 31, 60–62, 64, 70, 84, and 97 to identify associations with gene silencing and patient survival [45]. The highest negative correlation between methylation and gene expression and the strongest association with OS were found for CpGs 22, 25, 31, and 84. Growing evidence suggests that in particular CpGs in the intrapromoter enhancer are of regulatory and/or clinical relevance [46,47,48,49,50,51]. By using primer sequences from the literature [31], we targeted CpGs 72–83, with CpGs 82–83 being located in the intrapromoter enhancer. In 19 GBM samples and the IDH mutated sample GBMm01, the CpGs were unmethylated, in 19 GBM samples, the gliosarcoma sample GS01, and in T98G, they were methylated. *MGMT* promoter methylation has already been reported for gliosarcoma, but it seems to occur with a lower frequency than in GBM [52]. In most samples with a methylated *MGMT* promoter, we found the twelve CpGs rather heterogeneously methylated. Only four samples showed a significant difference between the methylation levels of CpGs 82–83 in the intrapromoter enhancer and those of CpGs 72–81, with CpGs 82–83 being more highly methylated. The methylation levels of CpGs 72–83 correlated strongly positively with each other, highlighting the relevance of DNA methylation in the promoter region. MGMT protein expression was only detected for samples with an unmethylated *MGMT* promoter (and for T98G, as described above). In samples that did not express the MGMT protein, the methylation levels were quite diverse, ranging from low to high methylation, indicating that even a low DNA methylation status of the promoter is sufficient to suppress MGMT expression. 

Cut-off values for stratifying the GBM samples by their *MGMT* promoter methylation status are critically discussed in the literature [17,53]. For *MGMT* promoter methylation levels determined by PSQ, the cut-off value is most commonly set at 8% or 9% [54,55,56]. By using a cut-off of 8%, OS was significantly higher for patients with a methylated *MGMT* promoter than for those with an unmethylated *MGMT* promoter which is in line with previous studies [14,15]. GBM01 was the only sample that did not express MGMT although the *MGMT* promoter was unmethylated. With 7.4 months, the OS of patient GBM01 was rather short, corresponding to that of other patients with an unmethylated *MGMT* promoter that did express MGMT. 

With respect to the enhancers investigated, we obtained the most interesting results for intergenic enhancer 2, located 560 kb upstream of the *MGMT* promoter, and intragenic enhancer 4, located in intron 2. Chen et al., who identified enhancer 2, found that activation of the enhancer positively correlated with MGMT expression [30]. By activating the enhancer in cell lines with low MGMT expression, MGMT expression increased, whereas deletion of the enhancer region in cell lines with high MGMT expression resulted in a drastic reduction in MGMT expression. By investigating the enhancer region in more detail, Chen et al. distinguished between two regions: “Del 1” at the 5’ end of the enhancer, and “Del 2” at the 3’ end and thus closer to the *MGMT* promoter. Deletion of “Del 1” resulted in a dramatic reduction in MGMT expression, whereas deletion of “Del 2” did not lead to a significant change in MGMT expression. When we developed assays for DNA methylation analysis of enhancer 2, we attempted to target both “Del 1”, containing CpGs 01–25, and “Del 2”, containing CpGs 26–46 of the enhancer. Finally, we established four assays (A–D), targeting 19 CpGs in total, with CpGs 05–08 (assay A) and CpGs 11–18 (assay B) being part of “Del 1”, CpGs 37–39 (assay D) being part of “Del 2”, and CpGs 24–27 (assay C) being located at the border of “Del 1” and “Del 2”. CpGs 05–08, CpGs 11–14, and CpGs 37–39 showed similar methylation levels in the glioma samples investigated. However, CpGs 15–18 were significantly more highly methylated, and CpGs 24–27 were significantly less methylated. In most glioma samples, the CpGs in enhancer 2 were methylated, with the exception of CpGs 24–27 in nine samples and CpGs 37–39 in one sample(s). The methylation levels of CpGs 15–18 strongly correlated with each other, and a strong positive correlation was also found for CpGs 37–39, independent of whether the samples were stratified by their *MGMT* promoter methylation status or not. We did not find significant differences in enhancer 2 methylation levels between the MGMT-expressing and -non-expressing samples, neither for the mean methylation of the target regions nor for the individual CpGs. However, we found a significant negative correlation between the mean methylation of all CpGs targeted in enhancer 2 and the MGMT protein levels for the GBM samples expressing MGMT. The mean methylation of CpGs 5–8, 11–14, and 37–39 was also negatively correlated with MGMT expression levels. A significant negative correlation was also found for individual CpGs 05, 06, 07, 08, 11, 12, 13, 14, 37, and 39. Our results suggest that the DNA methylation levels of CpGs in both enhancer 2 regions, “Del 1” and “Del 2” [30], have an impact on enhancer activity and consequently on MGMT protein expression, whereas the methylation status of CpGs at the border of the two regions does not seem to play a role. 

Application of four assays (A–D) developed in-house for the analysis of CpGs 01–03, CpGs 07–08, CpGs 09–13, and CpGs 19–22, respectively, in intragenic enhancer 4 (hs696) revealed that CpGs 19–22 were significantly more highly methylated in most samples than CpGs 01–03, CpGs 07–08, and CpGs 09–13 (*p* < 0.001). Interestingly, several samples exclusively with a methylated *MGMT* promoter showed low methylation (mean methylation < 25.0%) of CpGs 01–03, CpGs 07–08, and CpG 09–13, whereas high methylation (mean methylation > 75.0%) was found exclusively for samples with a non-methylated *MGMT* promoter. The methylation levels of the CpGs located in enhancer 4 correlated positively with each other. Most correlations were found when the samples were not stratified by their *MGMT* promoter methylation status. In addition, when the samples were not stratified, the methylation levels of almost all CpGs in enhancer 4 negatively correlated with the methylation levels of almost all CpGs in the *MGMT* promoter. The strongest negative correlations were found for CpG 03 and CpG 09 in enhancer 4. The methylation levels of enhancer 4 were significantly higher in the MGMT-protein-expressing GBM samples than in the -non-expressing GBM samples. This holds true not only for the mean methylation of the CpGs targeted by one and the same assay (assay A–D) but also for all individual CpGs. All our findings suggest that DNA methylation of enhancer 4 is associated with *MGMT* promoter methylation and *MGMT* protein expression. In addition, we found DNA methylation of CpGs 01–03 and CpGs 09–13 to be associated with OS. Patients in which these CpGs were methylated to a lower degree had longer OS than patients with a higher methylation status. Moreover, for patients with shorter OS, prediction based on the methylation status of enhancer 4 was more precise than prediction based on the promoter methylation status. 

By applying in-house assays for intergenic enhancers 1 (hs737) and 3 (hs699), we found strong positive correlations between CpGs located in the respective enhancers. In the case of both enhancers, correlations were stronger for samples with an unmethylated *MGMT* promoter. For samples with an unmethylated *MGMT* promoter, we also found strong positive correlations between the methylation levels of CpGs located in enhancer 1 and those located in enhancer 3. Methylation levels of four individual CpGs (CpGs 17, 18, 21, and 22) in enhancer 3 and the mean of all CpGs targeted were significantly higher in the MGMT-expressing GBM samples than in the -non-expressing GBM samples; for the CpGs in enhancer 1, we did not find a difference between the MGMT-expressing and -non-expressing samples. Neither for enhancer 1 nor for enhancer 3 did we find the methylation status to be associated with the OS of the patients. 

## 5. Conclusions

We have provided DNA methylation levels for four *MGMT* enhancers that have not been subjected to DNA methylation analysis before. Our findings indicate that the methylation levels of the CpGs in an enhancer located in *MGMT* intron 2 (enhancer 4, hs696) is significantly negatively correlated with the methylation levels of the CpGs in the *MGMT* promoter. CpGs of enhancer 4 were significantly more highly methylated in the MGMT-expressing samples compared to the -non-expressing samples. Moreover, low methylation of CpGs 01–03 and CpGs 09–13 of enhancer 4 turned out to be favorable for the OS of GBM patients. For the CpGs in enhancer 2, located 560 kb upstream of the *MGMT* promoter, we found a significantly negative correlation between methylation status and MGMT protein levels for GBM samples with an unmethylated *MGMT* promoter, expressing the MGMT protein. The methylation levels of four CpGs (CpG 17, 18, 21, and 22) of enhancer 3 (hs699), located upstream of the *MGMT* promoter and downstream of enhancer 2, were significantly higher in the MGMT-expressing GBM samples than in the -non-expressing GBM samples. The methylation status of enhancer 1 (hs737), located upstream and at a greatest distance from the *MGMT* promoter, was neither associated with *MGMT* promoter methylation or MGMT expression nor with the OS of GBM patients.

Our findings suggest that enhancer methylation contributes to MGMT regulation and is a potential prognosticator for GBM patient survival. However, it remains to be elucidated if DNA methylation of the four enhancers targeted is associated with SNP genotypes, e.g., *MGMT* rs16906252, and/or clinicopathological characteristics, including the proliferation marker Ki-67, progression-free survival, and the response to TMZ in GBM.

## Figures and Tables

**Figure 1 cells-12-01639-f001:**
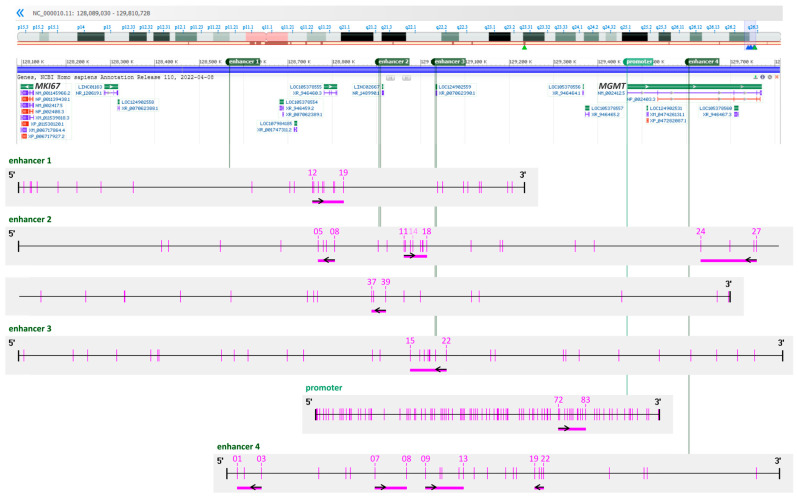
Schematic representation of the *MGMT* enhancer and promoter regions targeted. Enhancer 1 (hs737): 27 CpGs, 1138 bp, enhancer 2 (Chen et al. [30], 46 CpGs, 3313 bp, “Del 1”: CpGs 1–25, “Del 2”: CpGs 26–46; enhancer 3 (hs699): 33 CpGs, 1719 bp; enhancer 4 (hs696): 26 CpGs, 1243 bp; *MGMT* promoter: CpG island of 98 CpGs, 762 bp. Pink vertical lines indicate CpG positions, and pink horizontal bars CpGs targeted by the respective PSQ assay. Small black arrows refer to the sequencing direction related to the upper strand. The CpGs targeted are numbered according to their position in the respective enhancer/promoter. The representation of chromosome 10 including the *MGMT* gene location was taken from NCBI Genome Data Viewer [34]. The CpG line schemes were generated using Methyl Primer Express Software v1.0 (Thermo Fisher Scientific) and adapted manually.

**Figure 2 cells-12-01639-f002:**
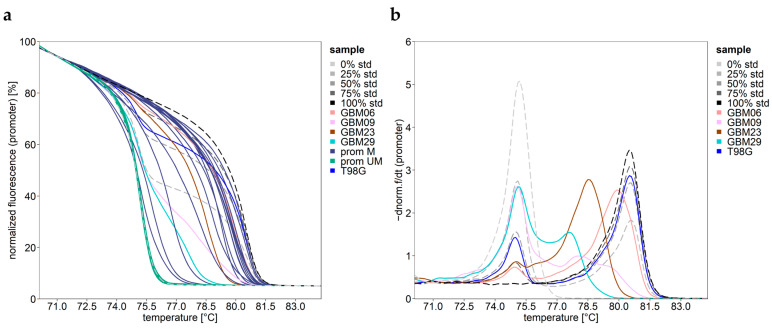
HRM curves for the *MGMT* promoter region. (**a**) Normalized melting plot. Samples with unmethylated promoter (green), methylated promoter yielding typical (purple) and melting curves with multiple melting transitions (differently colored). Non-methylated (~0%), methylated DNA (~100%), and their 25%, 50%, and 75% mixtures (dashed lines ranging from light gray (0%) to black (100%)). (**b**) Negative derivative of normalized melting curves for samples GBM06, GMB09, GBM23, GBM29, and T98G, indicating multiple sharp melting transitions.

**Figure 3 cells-12-01639-f003:**
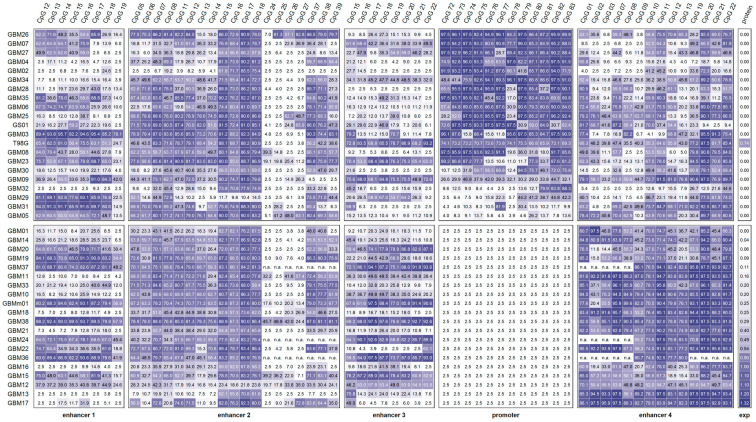
Heatmap for *MGMT* enhancer and promoter methylation status and MGMT expression in glioma. The heatmap is sorted by expression (exp) and mean promoter methylation. The upper panel shows samples with promoter (CpGs 72–83) methylation; the lower panel shows those with an unmethylated promoter. Tile labels: methylation level (mean of two independent PSQ runs), MGMT expression level: relative MGMT protein expression, determined by Western blot analysis, is given as the ratio to the MGMT overexpressing glioblastoma cell line GL80 [39]; n.a.: not analyzed due to limited sample amount. Color scale white 0%–purple 100%. GBM01–38: IDH-wild-type GBM, GBMm01: IDH1-mutated GBM, GS01 gliosarcoma patient, and T98G: commercial GBM cell line.

**Figure 4 cells-12-01639-f004:**
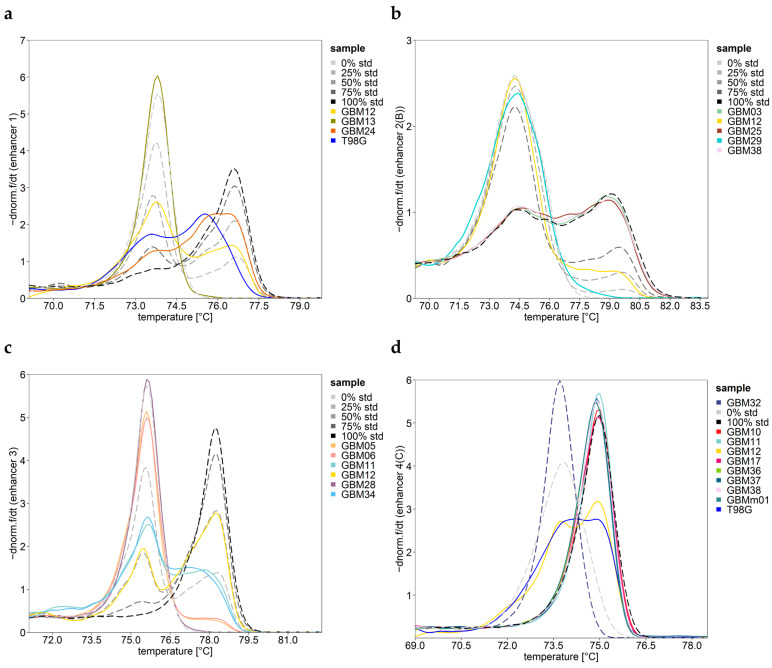
Negative derivative of normalized HRM curves for *MGMT* enhancer regions for samples being either non- or completely methylated or showing characteristic multiple melting transitions: (**a**) enhancer 1, (**b**) enhancer 2 (shown for assay B), (**c**) enhancer 3, (**d**) and enhancer 4 (shown for assay C). The PCR products of GBM12 (yellow) and T98G (blue) had two melting transitions in all four and two enhancers, respectively. DNA standards: non-methylated (~0%), methylated DNA (~100%), and 25%, 50%, and 75% mixtures thereof (dashed lines ranging from light gray (0%) to black (100%)). In the target regions of enhancer 4, the non-methylated standard turned out to be partially methylated (**d**). One representative measurement of two independent runs is shown. GBM01–38: IDH-wild-type GBM, GBMm01: IDH1-mutated GBM, GS01 gliosarcoma, and T98G: commercial GBM cell line.

**Figure 5 cells-12-01639-f005:**
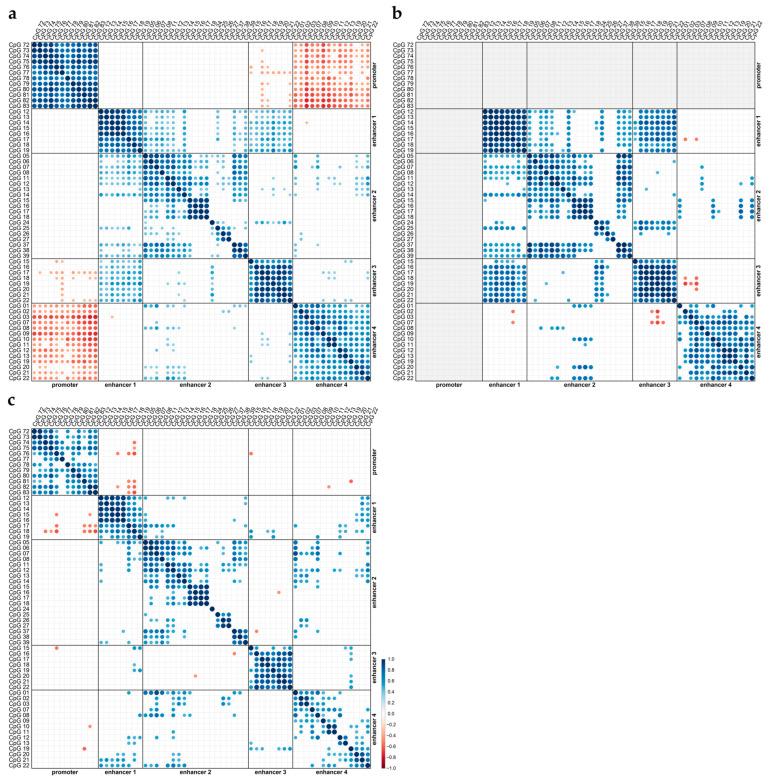
Correlations between methylation levels (mean of two independent runs) of individual CpGs of the *MGMT* promoter and enhancer regions. Correlation plots showing significant Pearson´s correlation coefficients (color range dark red (−1.0)–dark blue (1.0); point size (0–±1.0)), light gray background: not analyzable): (**a**) for all, (**b**) promoter unmethylated, and (**c**) promoter methylated IDH-wildtype GBM samples (GBM01–38). Pairwise complete comparisons were performed.

**Figure 6 cells-12-01639-f006:**
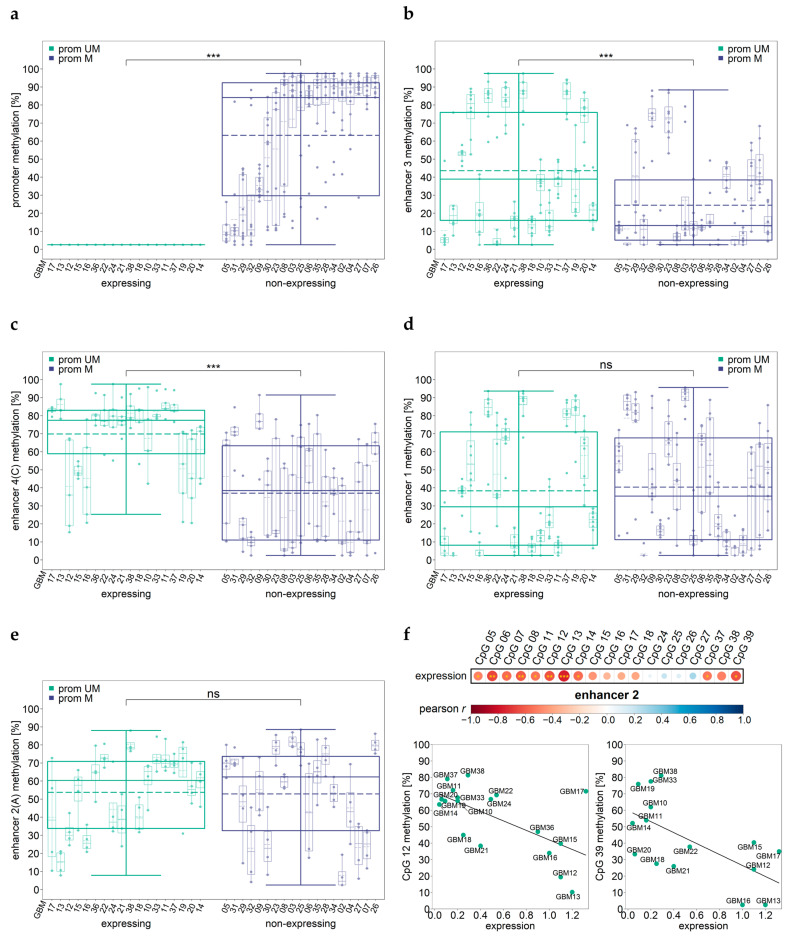
DNA methylation levels of the promoter and enhancer regions and MGMT protein levels. (**a**–**e**) Samples are ordered by increasing mean promoter methylation level and increasing MGMT expression. Outer boxplots show differences in methylation levels for the MGMT-expressing and -non-expressing samples (promoter unmethylated (green) and promoter methylated (purple)). Inner boxplots represent methylation variability by individual CpGs in one sample in the (**a**) promoter, (**b**) enhancer 3, (**c**) enhancer 4 (shown for assay C), (**d**) enhancer 1, and (**e**) enhancer 4 (shown for assay A) regions. (**f**) Correlation plots and two exemplary scatterplots for enhancer 2 for the unmethylated promoter samples are shown. Pearson´s correlation coefficients are shown as the color range dark red (−1.0)–dark blue (1.0); point size (0–±1.0); pairwise complete comparisons were performed; significant coefficients are highlighted in yellow (* *p* ≤ 0.05, ** *p* ≤ 0.01, *** *p* ≤ 0.001). Data points represent the mean of two independent runs. IDH-wild-type GBM samples (GBM02–38) were analyzed.

**Figure 7 cells-12-01639-f007:**
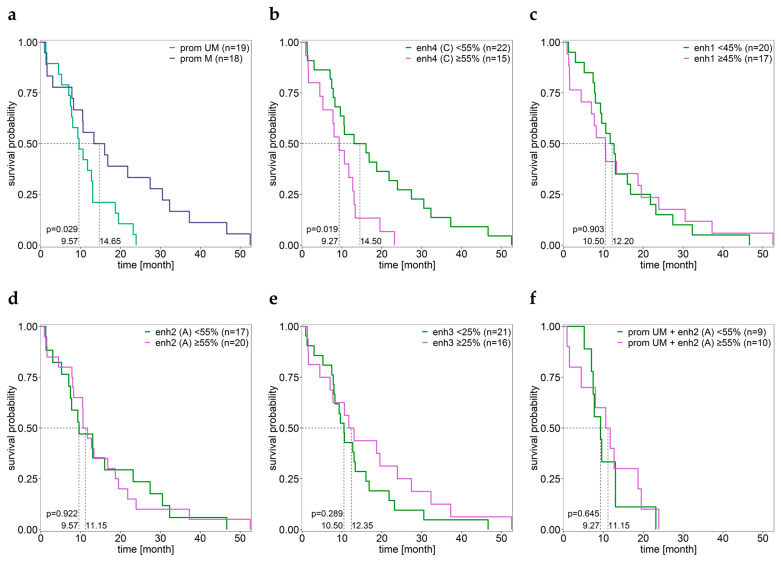
Kaplan Meier survival analysis. The IDH-wild-type GBM samples (GBM01–38) were grouped by their methylation status of the (**a**) promoter, (**b**) enhancer 4 (shown for assay C), (**c**) enhancer 1, (**d**) enhancer 2 (shown for assay A), and (**e**) enhancer 3 regions as well as (**f**) of enhancer 2 (shown for assay A) for promoter unmethylated samples. Patient GBM27 was excluded due to a lack of OS data.

**Table 1 cells-12-01639-t001:** Overview of PSQ assays.

Region	Primer Set	Primer Sequence (5′→3′)	Length [bp]	CpGs Analyzed
enhancer 1	A	F: AGAATGTAGATTTGGGATTAGTTAAT	229	12–19
(hs737)		R: [Btn] TAAAATACAAAATATACCCTCCAACA		
27 CpGs		S: TATATAAAGAAGGTTGGT		
enhancer 2	A	F: AGTTAGGAAATTAGAAATGGAATGTTT	255	05–08
(Chen et al.)		R: [Btn] CAAATCACACTCTAAATTCCCAATT		
46 CpGs		S: TGGTATTAGAGGTTA		
	B	F: [Btn] TTAAATAAGTGGTTTAGGTAGAGG	137	11–18
		R: TACTAAACATTCCATTTCTAATTTCC		
		S: CCATTTCTAATTTCCTAACTC		
	C	F: GTTGTAGGGTATATGAGTTTAGAT	271	24–27
		R: [Btn] TTCATAACTCAAATTAACACACACT		
		S: TTTTGTGTTGAATGG		
	D	F: GAGGTTATTTGGAAAGTTGAGAT	286	37–39
		R: [Btn] CTAATAATCCAAACCCTCTATTC		
		S: TTTAGTGTTATGGGAG		
enhancer 3	A	F: TGTGTTAGTTTTAGTGGTTTAGA	138	15–22
(hs699)		R: [Btn] TAACACACAAACCAATCTCTC		
33 CpGs		S: TAGTTTTAGTGGTTTAGAAGT		
enhancer 4	A	F: [Btn] GGAATGTGTTATTTAATTGGTATGT	204	01–03
(hs696)		R: CAAATCCCACAACAAATCCTTAT		
26 CpGs		S: TCAAAAAAAAAAAATCACC		
	B	F: GAGGTTTGATATAAGTAATGATGG	131	07–08
		R: [Btn] CCTCCTAATCCCACAATACAA		
		S: TAAGTAATGATGGTATG		
	C	F: AGGTTTGATATAAGTAATGATGGTAT	257	09–13
		R: [Btn] CRTATTCTCTCCCACTTCAATA		
		S: GTATTGTGGGATTAGGA		
	D	F: [Btn] GTGTATTGAAGTGGGAGAGAATA	241	19–22
		R: CAATAACAATTTTACAAACACAAATAACTT		
		S: ATAACTTTTCATTCA		
promoter	A [31]	F: GGATATGTTGGGATAGTT	98	72–83
98 CpGs		R: [Btn] CCCAAACACTCACCAAAT		
		S: GGATATGTTGGGATAGTT		

[Btn]: biotin, length: PCR product length; bp: base pairs, F: forward primer, R: reverse primer, and S: sequencing primer.

## Data Availability

The datasets generated during the current study are available from the corresponding author on reasonable request.

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
