# Peer review of "Association between MGMT Enhancer Methylation and MGMT Promoter Methylation, MGMT Protein Expression, and Overall Survival in Glioblastoma"

_cells, 2023, doi:10.3390/cells12121639_

Round 1

Reviewer 1 Report

The role of MGMT promoter methylation in GBM has been known for decades with variable correlation with survival and temozolomide sensitivity. This paper analyzed the methylation status of 4 MGMT enhancers, MGMT promoter, and MGMT protein expression and concluded that higher methylation levels of MGMT enhancers negatively correlated with MGMT promoter methylation in GBM patients and survival. A similarly negative correlation was found for the methylation of CpGs in enhancer 2 and of MGMT promoter in GBM patients with MGMT promoter methylation. 

Some minor modifications are suggested: 

In Conclusions, it is stated that the methylation status of Enhancer 2 correlates to MGMT promoter methylation, protein expression, and survival. This needs to be further clarified (Figures 5a, 6 and 7). The assessment of the correlations of enhancers 1 and 3 would be of benefit in the Conclusions Section.  

The presentation of Figure 1 was truncated and the reviewer could not view the entire figure. Please revise. Are the numbers methylation level or protein expression level? Please specify. 

Figures 6 and 7: The panels may be rearranged in order of enhancers #1-3, promoter, and enhancer #4 to be consistent with Figure 1. 

Add a summary table or heatmap for all GBM cases: Case number, promoter methylation status, protein expression level, survival time. 

Add a graphical or text summary on how the enhancers correlate with each other and with the MGMT promoter based on their findings. 

Sections 3.1.2-3.1.5 

The HRM analyses are complex. Can the authors give a brief summary to describe the major findings in each enhancer? How would these findings inform the next experiments/findings? 

Reviewer 2 Report

The manuscript “Association of MGMT Enhancer Methylation with MGMT Promoter Methylation, MGMT Protein Expression, and Overall Survival in Glioblastoma” is interesting and important in the research related to the poor medical prognosis of GBM. 

Some minor suggestions:

  1. A brief explanation in the Introduction on the relevance of DNA methylation could be interesting for non-expert readers (on L42-43?).
  2. The authors used 4 enhancers for their study, could they recommend the potential clinical value of testing these enhancers in individual cases?
  3. Some acronyms (e.g. OS) could be more explicit without abbreviation (e.g. overall survival, or survival, alone).

Reviewer 3 Report

The paper  entitled "Association of MGMT Enhancer Methylation with MGMT Promoter Methylation, MGMT Protein Expression, and Overall  Survival in Glioblastoma"  that I've finished to review, I  can conclude that it is a very good paper. Methods and results are clear and of impact, maybe the discussion could include recent literature regarding glioblastoma for example A Alafandi et al 2023.  Anyway, it is a good job.

Best regards
